analytical chemistry/spectroscopy/green chemistry

green analysis, dantrolene sodium, reduction, spectrofluorimetry, pharmaceutical formulations, content uniformity testing

**Author for correspondence:**
Nora A. Abdalah
e-mail: noraabdallah91@mans.edu.eg

This article has been edited by the Royal Society of Chemistry, including the commissioning, peer review process and editorial aspects up to the point of acceptance.

# Green spectrofluorimetric assay of dantrolene sodium via reduction method: application to content uniformity testing

Nora A. Abdalah, Mona E. Fathy, Manar M. Tolba, Amina M. El-Brashy and Fawzia A. Ibrahim

Department of Pharmaceutical Analytical Chemistry, Faculty of Pharmacy, Mansoura University, 35516 Mansoura, Egypt

Green analysis has turned out to be of a great value in all areas, including pharmaceutical analysis. Thus, it is extremely important to consider the environmental influence of each step in developing any analysis technique. The present work illustrates a validated, simple and green spectrofluorimetric method for the analysis of dantrolene sodium (DAN). The developed process is characterized by being of high sensitivity as well as being relatively inexpensive. The suggested technique based on the formation of a highly fluorescent product of DAN via its reduction by the aid of Zn/HCl system. The resulting fluorophore showed a powerful fluorescence at $\lambda_{em}$ 344 nm after excitation at $\lambda_{ex}$ 279 nm. Calibration graph revealed a great linear regression ($r = 0.9998$) within concentration ranging from 0.05 to 2.0 µg ml$^{-1}$. The suggested method had very low detection and quantification limits of 0.010 and 0.031 µg ml$^{-1}$, respectively. The applied technique was effectively used in the determination of DAN in its pharmaceutical preparations. The results were compared with those from using the official United States Pharmacopeia (USP) method and they were in a good agreement. Moreover, content uniformity testing of DAN in capsules was performed adopting the investigated technique with satisfying results. The greenness of the suggested technique was confirmed by the three standard assessment tools. Therefore, the developed technique can be used in the routine quality control analysis of DAN with minimum harmful impact on nature or individuals.

# 1. Introduction

Dantrolene sodium (DAN) is chemically designated as sodium;3-[(E)-[5-(4-nitrophenyl)furan-2-yl]methylideneamino]-2-oxo-4H-imidazol-5-olate (scheme 1) [1]. It is used to treat muscle tightness and cramping caused by certain nerve disorders such as injuries of the spinal cord, stroke and cerebral palsy. It works by relaxing the muscles, so it helps to reduce muscle pain and stiffness and improves the patient's ability to move around. DAN is also used with other therapies to prevent or treat special cases of high fever (malignant hyperthermia) related to anaesthesia and surgery [2]. It is official in the US [3], British [4] and Japanese [5] Pharmacopeias, where it was determined by HPLC. Several reported methods were published for DAN determination including spectrofluorimetry [6], HPLC [7–11], HPTLC [12], UPLC [13], voltammetry [14–17] and colorimetry [18]. Investigating the literature revealed that a few analytical techniques were presented for the analysis of DAN. Most of the reported methods were chromatographic methods, which usually require high-priced equipment and advanced skills to manipulate them successfully. Also, most of the reported analytical methods were of high hazardous impact on both the analyst and nature [6,8–10,12]. Reviewing the literature showed that there was no spectrofluorimetric method stated for DAN determination except for that published in 1973 [6].

The recommended technique in this research is based on the fact that DAN does not have native fluorescence, which is attributed to the presence of a nitro group attached to the aromatic ring in its structure. It is known that nitro group has a powerful fluorescence quenching effect [19] because of its electron withdrawing ability. However, once the amino group comes out from reducing the nitro group, DAN gains its remarkable intrinsic fluorescence [20]. Based on the obtained high fluorescence of DAN, this study was devoted to developing a highly sensitive spectrofluorimetric procedure to determine DAN in pure form along with single and combined dosage forms. Content uniformity testing was also conducted to ensure consistence of DAN capsules. Our target was also directed to assess the greenness of the suggested technique in order to prove its safety to the environment. Moreover, the suggested technique was extensively validated according to the ICH guidelines [21] providing an assurance of reliability of the method during normal use. Therefore, this method presents the advantages of sensitivity, greenness, lack of sophistication and cost-effectiveness. The method is also specific and selective because of the absence of interference encountered from capsule excipients or from co-formulated drug, paracetamol.

Green analytical chemistry (GAC) started appearing in the early 2000s [22,23]. This developing field cares about ensuring that the presented analytical procedures reduce consumption of harmful reagents and increase the protection of the analysts as well as the environment. [24]. Recently, there have been significant precautions in procedural means with the aim of avoiding or reducing the hazardous effect of analytical procedures. The main approaches involve re-using, replacement with greener options and decreasing the usage and detoxification of reagents and solvents. In this research, greener options were chosen in every step of the proposed method and its greenness was confirmed by the three standard assessment tools [25–27].

# 2. Experimental procedure

## 2.1. Apparatus

— For spectrofluorimetric measuring, a Cary Eclipse fluorescence spectrophotometer with a xenon flash lamp was used. The applied voltage was 600 V. Smoothing factor of 10 was applied to the obtained spectra. For manipulation of data, Cary Eclipse software from Agilent Technologies was used.
— A Consort pH-meter (Belgium) was used.
— A model SS101H230 Sonix ultrasonic bath (New York, USA).

## 2.2. Materials and reagents

— DAN was generously given by Chemipharm Pharmaceutical Industries (Giza Governorate, Egypt) with purity of 100.03% as stated by the official United States Pharmacopeia (USP) method [3].
— Dantrelax® capsules batch no. 200075A containing 25 mg of DAN are product of Chemipharm Pharmaceutical Industries and were bought from a neighbouring pharmacy in Egypt.

**Scheme 1.** Reduction of DAN to RDAN.

— Dantrelax compound® capsules batch no. 201123A containing 25 mg DAN and 200 mg paracetamol are product of Chemipharm Pharmaceutical Industries and were bought from a neighbouring pharmacy in Egypt.
— The water used throughout the processes was bidistilled water.
— All the solvents used were of HPLC grade while the reagents used were of analytical reagent grade.
— Surfactants such as 98% cetyltrimethylammonium bromide (CTAB), 94% sodium dodecyl sulfate (SDS), tween-80, carboxymethylcellulose (CMC) and 32% hydrochloric acid (HCl), and chemicals used for buffer preparations such as boric acid and sodium hydroxide were all obtained from El-Nasr Pharmaceutical Chemical Co. (ADWIC, Cairo, Egypt). All surfactants were used as 0.2% w/v aqueous solutions.
— Zn metal dust was bought from BDH Chemicals (Poole, UK).
— Acetonitrile, ethanol and n-propanol are Sigma-Aldrich products (Germany), while methanol was bought from Tedia (USA).
— Acetate and borate buffer covering the pH range of (3.0–5.0) and (6.0–10.0), respectively, were prepared at a concentration of 0.2 M. Britton Robinson Buffer (0.04 M) covering pH range of (2.0–7.5) was prepared according to USP guidelines [3].

## 2.3. Preparation of stock standard solution

An accurate amount of DAN pure powder was weighed, transferred to a 100 ml volumetric flask, and dissolved using ethanol as green diluting solvent to make stock solution of 500 µg ml$^{-1}$. The solution showed stability for at least 7 days if stored in the refrigerator.

## 2.4. Procedure for reduction of DAN

The used reduction system consisted of 32% HCl (1.0 ml) and Zn metal dust (0.4 g). DAN stock solution (5.0 ml) was added to a 50 ml conical flask containing 10 ml of bidistilled water. Next, the reduction system was added starting with HCl followed by the addition of Zn metal dust. The flask contents were left to interact for 20 min with intermittent shaking to guarantee the complete reduction, then filtered using a grade 1, Whatman filter paper, into a 100 ml volumetric flask. The 50 ml conical flask

was rinsed five times using 3 ml of water and the washings were passed through the used filter paper to the previous 100 ml measuring flask. After that, the same 100 ml volumetric flask was completed to the mark with distilled water to get working standard solution of 25.0 μg ml$^{-1}$ reduced dantrolene sodium (RDAN).

## 2.5. Calibration graph construction

Different volumes from RDAN working standard solution were transferred into a set of 10 ml volumetric flasks to get final concentrations in the range of 0.05–2.0 μg ml$^{-1}$. The flasks were completed to their mark with bidistilled water and then mixed. The fluorescence intensities of the obtained solutions were measured at 344 nm using 279 nm as excitation wavelength. The relative fluorescence intensities (RFI) were plotted versus the final RDAN concentrations (μg ml$^{-1}$) to get the calibration graph. Alternatively, the corresponding regression equation was obtained.

## 2.6. Analysis of DAN in single and combined dosage form

The content of 10 capsules of Dantrelax® 25 mg or Dantrelax Compound® were carefully evacuated, accurately weighed and blended. Then, an amount of the powder equivalent to one capsule of each dosage form was accurately weighed, transferred into two separate 100 ml measuring flasks followed by the addition of 60 ml of ethanol to each flask. The flask contents were then mixed by the aid of sonication for half an hour and completed to the mark with ethanol. Filtration of contents of each flask was done to get rid of the insoluble excipients. The filtrates were proceeded as explained in §§2.4. and 2.5. The capsules' nominal contents were calculated from the previously constructed calibration graph or by the corresponding regression equation.

## 2.7. Content uniformity testing for Dantrelax® 25 mg

Ten capsules of Dantrelax® (25 mg) were individually evacuated and transferred to 10 separate 100 ml measuring flasks followed by the addition of 60 ml of ethanol to each flask. The flask contents were then mixed by sonication for half an hour and completed to the mark with ethanol. The contents of each flask were filtered and the procedure for the reduction of DAN was then applied. Content uniformity testing was performed according to United States Pharmacopeia guidelines [3]. The acceptance value (AV) was then calculated after simultaneous analysis of 10 individual capsules.

# 3. Results and discussion

The proposed spectrofluorometric technique depended on conversion of DAN into a highly fluorescent product. This was done by the reduction of the aromatic nitro compound (DAN) using Zn/HCl system into the corresponding amino compound (RDAN) as shown in scheme 1. The RDAN showed powerful fluorescence due to the development of a stable resonance product of DAN by relation to former publications [20,28]. Consequently, RDAN was found to have a strong fluorescence at 279/344 nm. Figure 4 shows the fluorescence spectra of 1 μg ml$^{-1}$ DAN and RDAN at 279/344 nm. Several experimental parameters influencing the reduction process and fluorescence intensity of RDAN were thoroughly studied and optimized to achieve maximum fluorescence behaviour. Comparing with the only published spectrofluorimetric method [6], which used hazardous solvents such as dimethylformamide, the suggested technique was much greener with no harmful effect on the analyst or the environment. The greenness of the proposed method was well assessed adopting the three ideal assessment tools [25–27]. Moreover, the previously reported method did not include either the determination of DAN in pharmaceutical preparation or the testing of content uniformity of the capsule.

For more verification of the reliability and safety, the analytical performance of the recommended method was compared with some of the previously reported methods [6,8–10,12–15], as shown in table 1. The comparison shows the novelty, greenness, simplicity and high sensitivity of the proposed method.

**Table 1.** Analytical performance of the suggested technique and some reported methods.

| method | range | LOD | technique hazard and simplicity |
|---|---|---|---|
| spectrofluorimetry [6] | 0.1–4.0 µg ml$^{-1}$ | not available | It is a combination of extractions, chromatography and the fluorimetry. It is time consuming, as the extraction, evaporation and separation steps are performed in about 70 min or more [10]. Usage of dangerous solvents makes it a non-green method. |
| HPLC [8] | 1.0–10 µg ml$^{-1}$ | not available | Similar to the previous method, it involves prolonged extraction and separation steps. |
| HPLC [9,10] | 0.7–2.8 µg ml$^{-1}$ | 8 ng ml$^{-1}$ | It is very complicated and includes exhausting time-consuming extraction steps. Furthermore, the usage of many organic solvents has a very harmful effect on the analyst and nature. |
| HPTLC [12] | 0.1–1.5 µg band$^{-1}$ | 0.033 µg band$^{-1}$ | In terms of greenness, the usage of chloroform and ethyl acetate is not preferable owing to their huge toxicity and harmful effect on both the analyst and the environment. Also, this method is slightly time consuming as the saturation time is about $30 \pm 5$ min. |
| UPLC [13] | 0.5–10 µg ml$^{-1}$ | not available | This is a complicated technique through the usage of two columns and a photodiode array. The mobile phase contains acetonitrile which is considered one of the toxic solvents. |
| voltammetry [14] | 0.1–2.6 µg ml$^{-1}$ | not available | The used dropping mercury electrode (DME) is poisonous, so care should be taken in its handling. |
| DC and DP[a] polarographic methods [15] | $5 \times 10^5$ and $5 \times 10^6$ M respectively | not available | Dealing with DME and liquid nitrogen is very dangerous and needs high caution. |
| The proposed method in this research | 0.05–2 µg ml$^{-1}$ | 0.010 µg ml$^{-1}$ | Simple technique was employed. No toxic or hazardous solvents were used. This method can be applied in the pharmaceutical preparations and content uniformity testing. The degree of greenness was evaluated using three assessing tools. |

[a]DC, direct current; DP, differential pulse.

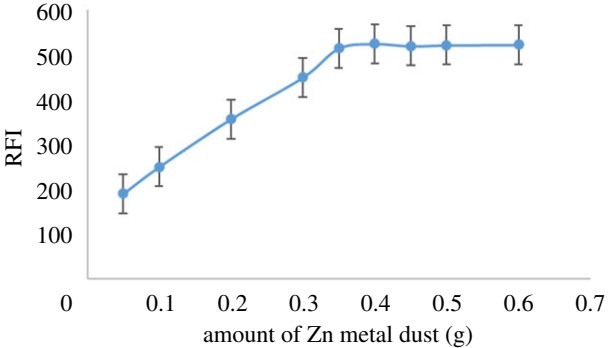

**Figure 1.** Influence of zinc metal dust amount on the formation of fluorophore RDAN (1.0 µg ml$^{-1}$).

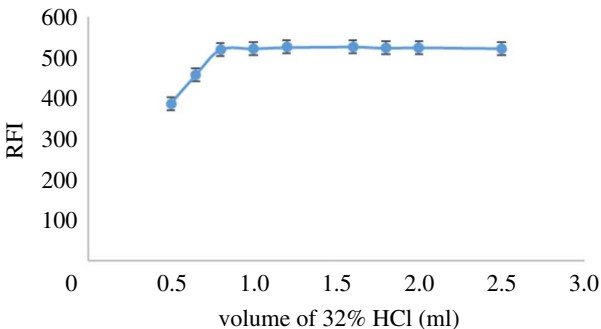

**Figure 2.** Influence of 32% HCl volume on the formation of the fluorophore RDAN (1.0 µg ml$^{-1}$).

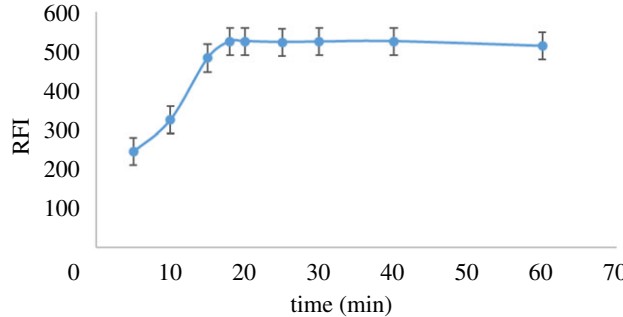

**Figure 3.** Influence of reaction time on the formation and stability of the formed fluorophore RDAN (1.0 µg ml$^{-1}$).

## 3.1. Experimental conditions optimization

### 3.1.1. Influence of the reduction system

The reduction system regarding Zn metal dust amount, HCl volume and the time of reduction at ambient temperature was studied to obtain the highest fluorescence intensity. The main target was to identify the most appropriate reduction system to obtain the greatest fluorescence intensity without consuming resources. The effect of Zn metal dust quantity from 0.05 to 0.7 g, volume of 32% HCl from 0.5 to 2.5 ml and reduction time from 5 to 60 min at normal temperature and pressure on the RFI were studied. It was noticed that RFI increased by increasing Zn metal dust and HCl up to 0.35 g of Zn metal dust and 0.8 ml of 32% HCl. Thus, 0.4 g of Zn metal dust and 1.0 ml of 32% HCl were the optimal amounts to ensure complete reduction of DAN (figures 1 and 2, respectively). Additionally, the influence of reduction time on the product fluorescent and its stability was studied. It was found that the reduction process started immediately, and the reaction product reached maximum intensity after 17 min (figure 3). The stability of the reduction product was checked and it was found that it did not change for at least 7 days. The fluorescence spectrum of the produced RDAN is illustrated in figure 4.

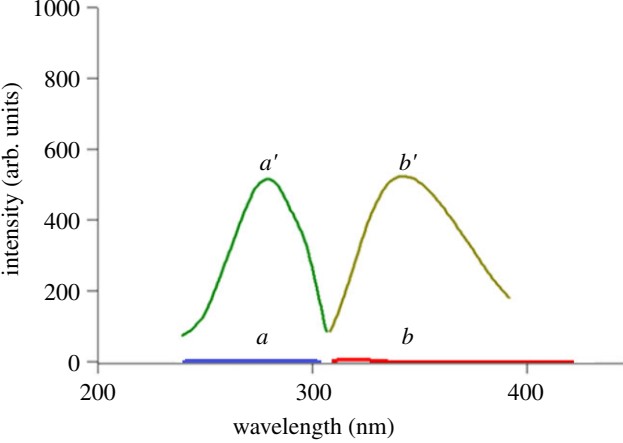

**Figure 4.** Fluorescence spectra of DAN and RDAN (1.0 µg ml$^{-1}$), where $a$ and $b$ are excitation spectra and $a'$ and $b'$ are emission spectra.

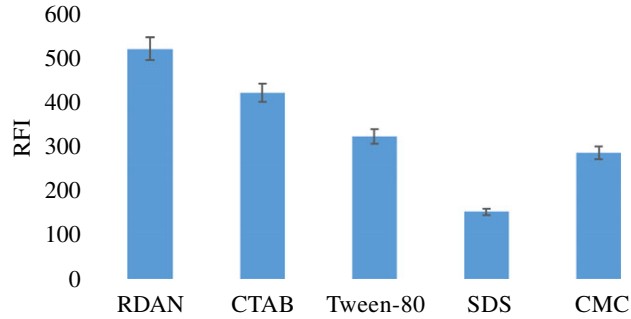

**Figure 5.** Influence of various organized media on RFI of 1.0 µg ml$^{-1}$ RDAN.

### 3.1.2. Influence of organized media

The effect of various organized media on the fluorescence characteristics of RDAN fluorophore were studied using cationic, anionic, non-ionic surfactants and macromolecules as (CTAB), (SDS), (Tween-80) and (CMC), respectively. One millilitre of each surfactant was added to a 10 ml measuring flask containing 1 ml (10 µg) RDAN solution, and the flask was completed with water. It was found that CTAB caused a slight decrease in the RFI of the RDAN while other tested organized media caused an obvious decrease in RDAN fluorescence as shown in figure 5.

### 3.1.3. Influence of pH

Different kinds of buffers, for example, 0.04 M Britton–Robinson buffers, 0.2 M of both acetate and borate buffer, covering pH range of 2–10 were applied to study the influence of pH on the fluorescence intensity of RDAN. The study revealed that the best RFI was achieved using water only without the need for any of the previously mentioned buffers, as demonstrated in figure 6. Consequently, no buffer was used during the experiment, since completing with only water produced the highest fluorescence intensity.

### 3.1.4. Influence of diluting solvent

The influence of several diluting solvents was studied to find out their impact on the RDAN fluorescence. The studied solvents were water, acetonitrile, methanol, ethanol and n-propanol. As shown in figure 7, it was discovered that bidistilled water was the most suitable solvent to be used, as it had the lowest blank reading and subsequently the highest RFI with reproducible results. Besides, water is the safest, less harmful, cost-effective and the ultimate green eco-friendly option as a solvent. Conversely, ethanol, acetonitrile and n-propanol caused a huge reduction in the fluorescence intensity of the RDAN.

## 3.2. Method validation

To consider the suggested technique a validated procedure, the ICH [21] guidelines was applied to check the validity of the suggested technique, and the results are summarized in table 2.

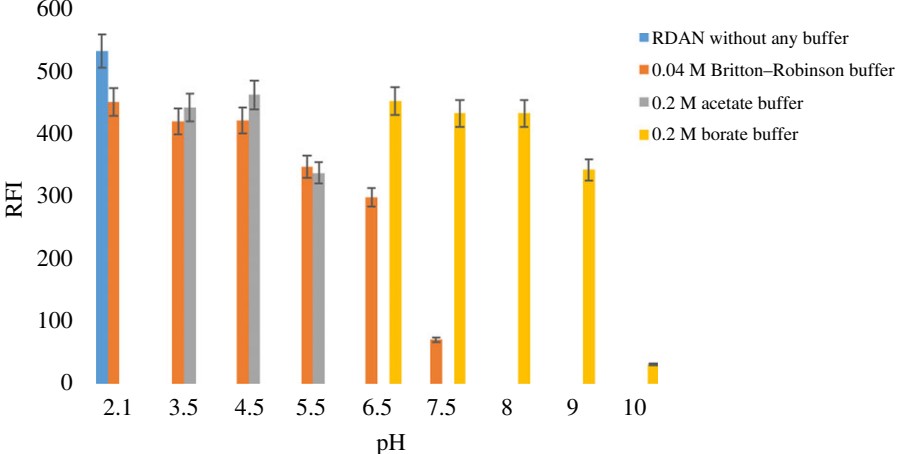

**Figure 6.** Influence of various buffers pH on RFI of 1.0 µg ml$^{-1}$ RDAN.

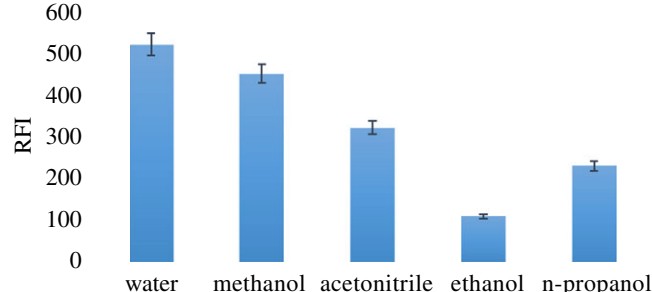

**Figure 7.** Influence of various diluting solvents on RFI of 1.0 µg ml$^{-1}$ RDAN.

**Table 2.** Analytical performance data by the suggested technique.

| parameter | | DAN |
|---|---|---|
| linearity range (µg ml$^{-1}$) | | 0.05–2.0 |
| intercept ($a$) | | 28.255 |
| slope ($b$) | | 488.712 |
| correlation coefficient ($r$) | | 0.9998 |
| s.d. of residuals ($S_{y/x}$) | | 6.153 |
| s.d. of intercept ($S_a$) | | 1.523 |
| s.d. of slope ($S_b$) | | 3.215 |
| percentage relative standard deviation, % RSD | | 1.610 |
| limit of detection, LOD (µg ml$^{-1}$) | | 0.010 |
| limit of quantitation, LOQ (µg ml$^{-1}$) | | 0.031 |
| Homoscedacticity[a] | calculated F | 1.17 |
| | tabulated F | 3.17 |

[a]Significance level 95% and degrees of freedom of nine for both numerator and denominator [29].

### 3.2.1. Linearity, detection and quantification limits

Calibration graph was constructed within the range of 0.05–2.0 µg ml$^{-1}$ of RDAN, each individual concentration relative fluorescence intensity was recorded three times followed by plotting it versus the corresponding concentration to obtain the calibration graph.

Table 2 contains a summary of the data statistical analysis [29]. The data shown proved the linearity of the calibration graph using the homoscedasticity test which showed $F$-test value of 1.17 while the tabulated $F$-value of 3.18. That ensures the homoscedasticity of the variance and the validity of the

**Table 3.** The results of DAN determination in pure form by the suggested technique and official USP method. N.B. The tabulated $t$ and $F$ values are 2.2 and 19.38, respectively, at $p = 0.05$ [29].

| parameter | suggested technique | Official USP method [3] |
|---|---|---|
| mean | 99.63 | 100.03 |
| ± s.d. | 1.60 | 1.08 |
| $t$ | 0.40 | |
| $F$ | 2.53 | |

**Table 4.** Precision data for DAN determination by the suggested technique. N.B. Each result is the average of three separate determinations.

| amount taken ($\mu$g ml$^{-1}$) | % found | % RSD | % error |
|---|---|---|---|
| intra-day | | | |
| 0.2 | 100.81 ± 0.77 | 0.77 | 0.44 |
| 0.5 | 99.89 ± 0.89 | 0.89 | 0.51 |
| 1.0 | 99.47 ± 0.20 | 0.21 | 0.12 |
| inter-day | | | |
| 0.2 | 100.15 ± 1.29 | 1.29 | 0.74 |
| 0.5 | 100.63 ± 1.43 | 1.42 | 0.82 |
| 1.0 | 99.57 ± 1.09 | 1.09 | 0.63 |

suggested technique in the studied range. Also, the developed method offers a very high sensitivity with very low detection and quantification limits. All these features are at a low price compared with other reported methods.

### 3.2.2. Accuracy and precision

There is no significant difference between the results obtained from the suggested technique and that from the official USP [3] regarding accuracy and precision (table 3). It is based on a liquid chromatographic analysis using reversed phase HPLC and gradient elution with different ratios of two different solutions; solution a (120 : 80 : 7 mixture of ammonium acetate buffer, acetonitrile and glacial acetic acid) and solution b (70 : 30 mixture of acetonitrile and water). The used column is C$_{18}$, the flow rate is 1.5 ml min$^{-1}$ and UV detection is at 365 nm. The comparison was performed using Student's $t$ and variance ratio $F$-tests [29].

Precision was assessed at two levels: repeatability was evaluated through analysis of three different concentrations of RDAN within the calibration curve three times each on the same day (intra-day precision) and intermediate precision by repeating analysis of every one of the three concentrations on three sequential days (inter-day precision). The two levels of precision were proven by the small relative standard deviations and are shown in table 4.

### 3.2.3. Specificity and selectivity

Thanks to the specificity of the suggested technique, the excipients of the capsule preparations did not affect the results. Moreover, paracetamol in the combined capsule showed no interference in the analysis of DAN which indicate the selectivity of the suggested technique.

### 3.2.4. Robustness

The suggested technique robustness was determined by the steadiness of the fluorescence intensity upon intended calculated slight alterations in the optimum selected conditions. The changed parameters included Zn metal dust, 32% HCl amounts (0.4 ± 0.05 g), (1.0 ± 0.2 ml), respectively and the reduction

**Table 5.** Robustness of the suggested technique using DAN (1.0 µg).

| parameter | mean ± s.d. | % RSD |
|---|---|---|
| amount of zinc metal dust, 0.35, 0.40 and 0.45 g | 100.72 ± 1.13 | 1.12 |
| volume of 32% HCl, 0.9, 1.0 and 1.1 ml | 99.81 ± 1.12 | 1.12 |
| reduction reaction time, 18,20 and 22 min | 100.07 ± 1.80 | 1.80 |

**Table 6.** Results for the DAN determination in single and co-formulated capsules by the suggested technique and official USP method. N.B. The tabulated $t$ and $F$ values are 2.77 and 19.0, respectively at $p = 0.05$ [29]. Dantrelax® or Dantrelax compound® capsules containing 25 mg of DAN or 25 mg DAN and 200 mg paracetamol, respectively. The nominal contents of the DAN in Dantrelax® and Dantrelax compound® capsules were found to be 24.98 and 24.97 mg, respectively.

| parameter | pharmaceutical dosage forms | | Official USP method [3] | |
|---|---|---|---|---|
| | Dantrelax® capsules | Dantrelax compound® capsules | Dantrelax® capsules | Dantrelax compound® capsules |
| mean | 99.93 | 99.89 | 100.30 | 100.03 |
| ± s.d. | 1.14 | 1.35 | 1.01 | 0.72 |
| $t$ | 0.11 | 0.16 | | |
| $F$ | 1.27 | 3.48 | | |

time (20 ± 2 min). These slight alterations that might happen through the experimental process had no effect on the RFI of RDAN, as demonstrated in table 5.

# 4. Application

## 4.1. Pharmaceutical applications

One of the targets of this study was to analyse DAN capsules either in single capsules (Dantrelax® capsules) or in compound capsules (Dantrelax compound® capsules). Comparing the obtained results with those of the official USP method [3] using $t$- and $F$-tests [29] showed no difference between them [3] concerning precision and accuracy, respectively as shown in table 6.

## 4.2. Content uniformity testing

As a result of the high sensitivity of the suggested technique and its capability of easily measuring the fluorescence intensity of a single capsule extract after reduction with satisfactory accuracy, the suggested technique is perfectly suitable for content uniformity testing of DAN in its capsules. The test steps were carried out referring to the USP method [3]. The AV was determined, and it came out to be less than the maximum allowed AV (L1). The results show an outstanding uniformity for DAN in its capsules as shown in table 7.

## 4.3. Greenness evaluation

Nowadays the greenness of an analytical method became as important as its sensitivity. The greenness of the analytical method guarantees protection of human and environmental factors from damage and unpleasant effect of chemicals. The authors developed the presented technique making sure that it fulfils as much as possible the greenness requirements. The analytical method is generally considered ultimate green if it totally avoids the use of toxic organic solvents, time-wasting derivatization procedures, energy utilization and waste production. There are currently some established standards for greenness evaluation of analytical methods, such as the National Environmental Methods Index (NEMI) labelling, analytical eco-scale score [25] and recently the Green Analytical Procedure Index

**Table 7.** Results of content uniformity testing of DAN in capsules using the suggested technique.

| parameter | percentage of the label claim<br>Dantrelax® capsules 25 mg of DAN/capsule |
|---|---|
| mean (x̄) | 100.28 |
| % RSD | 1.39 |
| % Error | 0.44 |
| calculated value (AV) | 3.34 |
| Max. allowed AV (L1) [3] | 15 |

**Table 8.** Results for greenness assessment of the suggested technique by three standard green analytical chemistry metric tools.

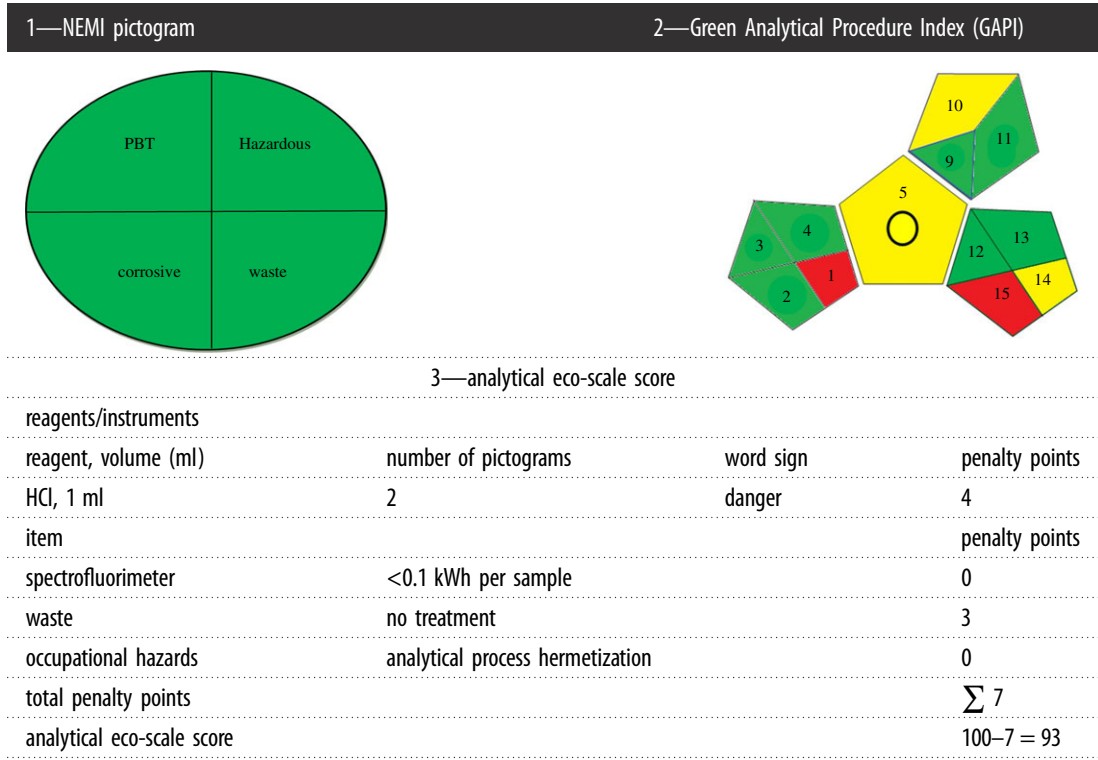

| 1—NEMI pictogram | | | 2—Green Analytical Procedure Index (GAPI) |
|---|---|---|---|

| 3—analytical eco-scale score | | | |
|---|---|---|---|
| reagents/instruments | | | |
| reagent, volume (ml) | number of pictograms | word sign | penalty points |
| HCl, 1 ml | 2 | danger | 4 |
| item | | | penalty points |
| spectrofluorimeter | <0.1 kWh per sample | | 0 |
| waste | no treatment | | 3 |
| occupational hazards | analytical process hermetization | | 0 |
| total penalty points | | | ∑ 7 |
| analytical eco-scale score | | | 100−7 = 93 |

(GAPI) [26]. The above three evaluation methods were applied to the suggested analytical technique to evaluate its greenness, as explained below.

The NEMI pictogram for the suggested analytical technique is demonstrated in table 8. The table reveals that the method meets all the criteria to be a 'green method'. Water and HCl used in the suggested technique are neither considered persistent bioaccumulative toxic (PBT) [30] nor hazardous [31] by the EPA's Toxic Release Inventory [32,33]. The waste volume is not more than 50 g or ml and the method pH is not less than 2 and not more than 12. That is, the four terms of the greenness profile are fulfilled in the proposed analytical method. So far, this assessment tool is qualitative only, as the quantity of reagents or the kind of hazards was not involved [25].

Analytical eco-scale is a semi-quantitative tool and depends on several parameters, such as the amount of reagents, hazards (as defined by the Globally Harmonized System (GHS) for labelling of chemicals), energy consumption, industrial hazard, waste amount and treatment way. As shown in table 8, penalty points are calculated according to these parameters [27]. The penalty points are then added for the whole process and included in the eco-scale calculation, using that equation:

$$\text{Analytical eco-scale} = 100 - \text{total penalty points}$$

**Table 9.** Green Analytical Procedure Index (GAPI) parameters for the suggested method.

| category[a] | description |
| --- | --- |
| sample preparation | |
| collection (1) | off-line |
| preservation (2) | none |
| transport (3) | none |
| storage (4) | normal condition |
| type of method: direct or indirect (5) | indirect (simple preparation(filtration)) |
| scale of extraction (6)[c] | — |
| solvents/reagents used (7) | — |
| additional treatments (8) | — |
| reagent and solvents | |
| amount (9) | <10 ml |
| health hazard (10) | HCl = 3 and Zn = 0) |
| safety hazard (11) | HCl = 0 and 1 Zn = 0 |
| instrumentation | |
| energy (12) | ≤1.0 kWh per sample |
| occupational hazard (13) | hermetic sealing of the analytical process |
| waste (14) | 10 ml |
| waste treatment (15) | no treatment |
| quantification[b] | yes |

[a]The numbers between parentheses are the numbers in the GAPI pictogram (on the right in table 8).

[b]The circle inside the central pentagram represent the quantification property of the method.

[c]The pentagram representing items number 6, 7, 8 is not shown in the GAPI pictogram since no sample preparation steps is included in the procedure.

The closer the results to 100, the greener the analytical method [27]. In the proposed technique, the score was 93 indicating an excellent green methodology.

Tables 8 and 9 illustrate that the proposed technique fulfilled the majority of GAPI [26] criteria except field 15 (coloured red) which describes the waste without treatment. Field 5 is coloured yellow as the method includes simple preparation (filtration), field 10 is coloured yellow as the reagents used have moderate health effects as stated by National Fire Protection Association (NFPA) [26], while field 14 is yellow because the waste produced is 10 ml per sample. Overall, the results indicate the greenness of the proposed technique.

In summary, the proposed technique is considered green based on the results of the three independent assessment techniques. The proposed technique is environmentally safe for the determination of DAN with minimal impact on human health. The greenness of the suggested technique complements its simplicity, rapidness and sensitivity.

# 5. Conclusion

This study presents and validates a novel green, sensitive, time-saving and cost-effective spectrofluorimetric technique for DAN determination by chemical reduction mechanism. The high sensitivity and selectivity of the proposed technique made it easy and appropriate for DAN determination in single and co-formulated pharmaceutical preparation. This method did not require complicated handling like that associated with other techniques, such as chromatography. Additionally, the usage of dangerous organic solvents was not essential, which distinguishes this technique to be of lower toxicity and cost. The content uniformity of DAN in the Dantrelax® capsules was also verified. Therefore, the presented technique could be suited for routine quality control in pharmaceutical manufacturing depending on its high sensitivity and greenness.

Data accessibility. Data available from the Dryad Digital Repository: https://doi.org/10.5061/dryad.gxd2547kv [34].

Authors' contributions. N.A.A. carried out the laboratory work, participated in data analysis and drafted the manuscript; M.E.F. and M.M.T. participated in the design of the study and carried out the statistical analyses; A.M.E.-B. and F.A.I. designed the study, coordinated the study and revised the manuscript. All authors approved the manuscript for publication.

Competing interests. We declare we have no competing interests.

Funding. No funding supported this research.

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
