## [Peer Review File · Royal Society Open Science]

Review History

RSOS-210562.R0 (Original submission)

Review form: Reviewer 1

Is the manuscript scientifically sound in its present form?

Yes

Are the interpretations and conclusions justified by the results?

Yes

Is the language acceptable?

Yes

Do you have any ethical concerns with this paper?

No

Have you any concerns about statistical analyses in this paper?

No

Recommendation?

Accept with minor revision (please list in comments)

Comments to the Author(s)

Comment 1:

The title does not well describe the content; in other words the title is very much generalized.

Comment 2:

In section 2.5. the sentence "Different volumes from RDAN working standard solution were moved into a set of 10 mL volumetric flask" should be changed to "Different volumes from RDAN working standard solution were transferred into a set of 10 mL volumetric flask"

Comment 3:

The paragraph under section "2.7. Content Uniformity testing for Danterelax® 25 mg" is not clear for the reader, I suggest paraphrasing of it.

Comment 4:

The apparatus used voltage should be mentioned.

Comment 5:

The authors should mention the reference method in detail.

Comment 6:

The authors should show the error bars in 6, 7 and 8 figures.

Comment 7:

References 29-31 seem to need correction.

Comment 8:

The nominal content of the capsules should be mentioned under table 6

Review form: Reviewer 2

Is the manuscript scientifically sound in its present form?

Yes

Are the interpretations and conclusions justified by the results?

Yes

Is the language acceptable?

Yes

Do you have any ethical concerns with this paper?

No

Have you any concerns about statistical analyses in this paper?

No

Recommendation?

Major revision is needed (please make suggestions in comments)

Comments to the Author(s)

Comments:

1. Page 3, lines 19 - 21: "The suggested method had a very low detection and quantification limit of 0.007 ng mL⁻¹ and 0.019 ng mL⁻¹, respectively." I think that this is a typing mistake because spectrofluorimetric can never reach this abnormally extreme low sensitivity. In addition, the linear range of the method started at 0.05 µg mL. Therefore, the correct words should be "0.007 µg mL⁻¹ and 0.019 µg mL⁻¹, respectively ". This correction must be made also in page 10,

line 10 and in table 1 (page 20) and in table 2 (Page 21). If the values of the slope (488.728) and intercept (28.234) are used to calculate the values of RFI that correspond for the LOD and LOQ concentrations it will be 0.0034 and 0.0093 . These values are far below the intercept itself.

2. Page 4, line 28-29: The Authors state that "That very old method [6] included the determination of DAN only in biological fluids". It is not a drawback of the method of being published at 1973. Meanwhile, the same technique is applied in the current method. In addition, the reported spectrofluorimetric method is superior compared to the current method of being applied to biological fluids. In such analysis, sample clean up procedure should be performed to extract the drug from the complex matrix of the biological sample and this is not considered a drawback of the method.

3. Page 8, line 10: the Authors state that " Since DAN on its own has no native fluorescence (Fig. 5a), the proposed spectrofluorometric " This statement is not correct because the reported method in reference [6] was based on measuring the native fluorescence of the drug. This sentence should be modified.

4. Page 11, section 4.3. Greenness evaluation: The authors should summarize this section and delete the well-known details.

5. Page 14, line 10: what is "RCNP

6. The structure of the studied drug presented in Figure 1, page 15, is repeated in scheme 1. So Figure 1 can be deleted.

7. Figure 5a in page 17 its title is "Figure 5a: Native fluorescence spectra of DAN (1.0 $\mu\text{g mL}^{-1}$). " However, the figure shows only excitation spectra of the drug as the band is around 279 nm . The emission band is not appeared at all. The authors should present for curves; two excitation spectra of the drug and its reduced form and two emission spectra for he drug and its reduced form. The concentration of the drug and its reduced form should be the same . This will show the effect of reduction on the fluorescence spectra.

8. Figure 5b in page 17 has no meaning because the third dimension is not clear (The vertical axis is for RFI values and the horizontal is for wavelength, So the third (perpendicular to the page) is standing for what??).

9. Figure 7 at page 18: There are 4 bars at pH 6.5, the blue bar is stand for the reduced drug without any buffer, However, the pH of this solution would be ≈ 2.0 but not 6.5 due to the presence of the remaining HCl from the reduction step. Acetate buffer cannot be used outside the pH range of 3.7 – 5.7 otherwise, it will lose its buffer action.

10. Page 21, table 2, the values of S.D for both slope and intercept are extremely low and should be re-calculated. Although, the method involved multi-steps including; reduction for 20 minute, filtration and completion with the solvent. These step should elevate the values of SD,

11. Table 1 page 20: The text in the last column should be deleted and moved to the manuscript. The table should include only numerical values.

12. Tables should be summarized and individual data should be deleted.

Decision letter (RSOS-210562.R0)

Dear Dr Abdallah:

Title: Green Spectrofluorimetric Method for the Determination of Dantrolene Sodium Alone or in Presence of Paracetamol

Manuscript ID: RSOS-210562

The editor assigned to your manuscript has now received comments from reviewers. We would like you to revise your paper in accordance with the referee and Subject Editor suggestions which can be found below (not including confidential reports to the Editor). Please note this decision does not guarantee eventual acceptance.

Please submit your revised paper before 04-Jun-2021. Please note that the revision deadline will expire at 00.00am on this date. If we do not hear from you within this time then it will be assumed that the paper has been withdrawn. In exceptional circumstances, extensions may be possible if agreed with the Editorial Office in advance. We do not allow multiple rounds of revision so we urge you to make every effort to fully address all of the comments at this stage. If deemed necessary by the Editors, your manuscript will be sent back to one or more of the original reviewers for assessment. If the original reviewers are not available we may invite new reviewers.

RSC Associate Editor:
Comments to the Author:
(There are no comments.)

RSC Subject Editor:
Comments to the Author:
(There are no comments.)

Reviewers' Comments to Author:

Reviewer: 1

Comments to the Author(s)

Comment 1:

The title does not well describe the content; in other words the title is very much generalized.

Comment 2:

In section 2.5. the sentence "Different volumes from RDAN working standard solution were moved into a set of 10 mL volumetric flask" should be changed to "Different volumes from RDAN working standard solution were transferred into a set of 10 mL volumetric flask"

Comment 3:

The paragraph under section "2.7. Content Uniformity testing for Danterelax® 25 mg" is not clear for the reader, I suggest paraphrasing of it.

Comment 4:

The apparatus used voltage should be mentioned.

Comment 5:

The authors should mention the reference method in detail.

Comment 6:

The authors should show the error bars in 6, 7 and 8 figures.

Comment 7:

References 29-31 seem to need correction.

Comment 8:

The nominal content of the capsules should be mentioned under table 6

Reviewer: 2

Comments to the Author(s)

Comments:

1. Page 3, lines 19 - 21: "The suggested method had a very low detection and quantification limit of 0.007 ng mL⁻¹ and 0.019 ng mL⁻¹, respectively." I think that this is a typing mistake because spectrofluorimetric can never reach this abnormally extreme low sensitivity. In addition, the linear range of the method started at 0.05 µg mL. Therefore, the correct words should be "0.007 µg mL⁻¹ and 0.019 µg mL⁻¹, respectively ". This correction must be made also in page 10, line 10 and in table 1 (page 20) and in table 2 (Page 21). If the values of the slope (488.728) and intercept (28.234) are used to calculate the values of RFI that correspond for the LOD and LOQ concentrations it will be 0.0034 and 0.0093 . These values are far below the intercept itself.

2. Page 4, line 28-29: The Authors state that "That very old method [6] included the determination of DAN only in biological fluids". It is not a drawback of the method of being published at 1973. Meanwhile, the same technique is applied in the current method. In addition, the reported spectrofluorimetric method is superior compared to the current method of being applied to biological fluids. In such analysis, sample clean up procedure should be performed to extract the drug from the complex matrix of the biological sample and this is not considered a drawback of the method.

3. Page 8, line 10: the Authors state that " Since DAN on its own has no native fluorescence (Fig. 5a), the proposed spectrofluorometric " This statement is not correct because the reported method in reference [6] was based on measuring the native fluorescence of the drug. This sentence should be modified.

4. Page 11, section 4.3. Greenness evaluation: The authors should summarize this section and delete the well-known details.

5. Page 14, line 10: what is "RCNP

6. The structure of the studied drug presented in Figure 1, page 15, is repeated in scheme 1. So Figure 1 can be deleted.

7. Figure 5a in page 17 its title is "Figure 5a: Native fluorescence spectra of DAN (1.0 $\mu\text{g mL}^{-1}$). " However, the figure shows only excitation spectra of the drug as the band is around 279 nm . The emission band is not appeared at all. The authors should present for curves; two excitation spectra of the drug and its reduced form and two emission spectra for he drug and its reduced form. The concentration of the drug and its reduced form should be the same . This will show the effect of reduction on the fluorecence spectra.
8. Figure 5b in page 17 has no meaning because the third dimension is not clear (The vertical axis is for RFI values and the horizontal is for wavelength, So the third (perpendicular to the page) is standing for what??).
9. Figure 7 at page 18: There are 4 bars at pH 6.5, the blue bar is stand for the reduced drug without any buffer, However, the pH of this solution would be ≈ 2.0 but not 6.5 due to the presence of the remaining HCl from the reduction step. Acetate buffer cannot be used outside the pH range of 3.7 - 5.7 otherwise, it will lose its buffer action.
10. Page 21, table 2, the values of S.D for both slope and intercept are extremely low and should be re-calculated. Although, the method involved multi-steps including; reduction for 20 minute, filtration and completion with the solvent. These step should elevate the values of SD,
11. Table 1 page 20: The text in the last column should be deleted and moved to the manuscript. The table should include only numerical values.
12. Tables should be summarized and individual data should be deleted.

Author's Response to Decision Letter for (RSOS-210562.R0)

See Appendix A.

RSOS-210562.R1 (Revision)

Review form: Reviewer 1

Is the manuscript scientifically sound in its present form?

Yes

Are the interpretations and conclusions justified by the results?

Yes

Is the language acceptable?

Yes

Do you have any ethical concerns with this paper?

No

Have you any concerns about statistical analyses in this paper?

No

Recommendation?

Accept as is

Comments to the Author(s)

All recommended corrections have been made. The manuscript could be accepted in its present format

Decision letter (RSOS-210562.R1)

Dear Dr Abdallah:

Title: Green Spectrofluorimetric Assay of Dantrolene Sodium via Reduction Method: Application to Content Uniformity Testing
Manuscript ID: RSOS-210562.R1

It is a pleasure to accept your manuscript in its current form for publication in Royal Society Open Science. The chemistry content of Royal Society Open Science is published in collaboration with the Royal Society of Chemistry.

RSC Associate Editor:
Comments to the Author:
(There are no comments.)

RSC Subject Editor:
Comments to the Author:
(There are no comments.)

Reviewer(s)' Comments to Author:

Reviewer: 1

Comments to the Author(s)

All recommended corrections have been made. The manuscript could be accepted in its present format

Appendix A

Response to Reviewers' Comments

The authors thank the editor and the reviewers for their thorough reviews and comments which were generally in target and helped improve the manuscript. All the comments were carefully considered in preparing the revised version and a point-to-point response for these comments are list in red below.

REVIEWER #1

Comment 1:

The title does not well describe the content; in other words, the title is very much generalized.

Response:

The title was modified in the revised manuscript as follows: "Green Spectrofluorimetric Assay of Dantrolene Sodium *via* Reduction Method: Application to Content Uniformity Testing"

Comment 2:

In section 2.5. the sentence "Different volumes from RDAN working standard solution were moved into a set of 10 mL volumetric flask" should be changed to "Different volumes from RDAN working standard solution were transferred into a set of 10 mL volumetric flask"

Response:

This sentence was changed in the revised manuscript, as recommended by the reviewer.

Comment 3:

The paragraph under section "2.7. Content Uniformity testing for Danterelax[®] 25 mg" is not clear for the reader, I suggest paraphrasing of it.

Response:

The mentioned paragraph was paraphrased in the revised manuscript, as recommended by the reviewer. The revised paragraph is: "Ten capsules of Danterelax[®] (25 mg) were individually evacuated and transferred to ten separate 100 mL measuring flasks followed by the addition of 60 mL of ethanol to each flask. The flasks contents were then mixed by sonication for half an hour and completed to the mark with ethanol.

The contents of each flask were filtered and the procedure for the reduction of DAN was then applied. Content uniformity testing was performed according to United States Pharmacopeia guidelines [3]. The acceptance value (AV) was then calculated after simultaneous analysis of 10 individual capsules”.

Comment 4:

The apparatus used voltage should be mentioned.

Response:

The applied voltage was included in the revised manuscript as recommended by the reviewer, as follows: “The applied voltage was 600 V.”

Comment 5:

The authors should mention the reference method in detail.

Response:

The reference method was mentioned in detail in the revised manuscript, as follows: “The reference method is based on a liquid chromatographic analysis using reversed phase HPLC and gradient elution with different ratios of two different solutions; solution a (120:80:7 mixture of ammonium acetate buffer, acetonitrile, and glacial acetic acid) and solution b (70:30 mixture of acetonitrile and water). The used column is C₁₈, the flow rate is 1.5 mL/min and UV detection is at 365 nm.”

Comment 6:

The authors should show the error bars in 6, 7 and 8 figures.

Response:

The error bars were added to old Figures 6, 7 and 8 (now Figures 5, 6 and 7) in the revised manuscript, as recommended by the reviewer.

Comment 7:

References 29-31 seem to need correction.

Response:

We thank the reviewer for pointing this typo to the authors. References 29-31 were edited in the revised manuscript, as follows:

29. Miller J, Miller JC. 2018 Statistics and chemometrics for analytical chemistry. New York, NY: Pearson Education.

30. Persistent Bioaccumulative Toxic (PBT) Chemicals Covered by the TRI Program, United States Environmental Protection Agency <https://www.epa.gov/toxicsrelease-inventory-tri-program/persistent-bioaccumulative-toxic-pbt-chemicalscovered-tri> (accessed 20 February 2021).

31. Hazardous Waste Listings, United States Environmental Protection Agency, 2012, https://www.epa.gov/sites/production/files/2016-01/documents/hw_listref_sep2012.pdf, Accessed date: 20 February 2021.

Comment 8:

The nominal content of the capsules should be mentioned under table 6

Response:

The nominal content of the capsules was mentioned under Table 6 in the revised manuscript, as recommended by the reviewer.

REVIEWER #2

1. Page 3, lines 19 – 21: "The suggested method had a very low detection and quantification limit of 0.007 ng mL⁻¹ and 0.019 ng mL⁻¹, respectively." I think that this is a typing mistake because spectrofluorimetric can never reach this abnormally extreme low sensitivity. In addition, the linear range of the method started at 0.05 µg mL. Therefore, the correct words should be "0.007 µg mL⁻¹ and 0.019 µg mL⁻¹, respectively ". This correction must be made also in page 10, line 10 and in table 1 (page 20) and in table 2 (Page 21). If the values of the slope (488.728) and intercept (28.234) are used to calculate the values of RFI that correspond for the LOD and LOQ concentrations it will be 0.0034 and 0.0093. These values are far below the intercept itself.

Response:

The authors thank the reviewer for pointing out that typing mistake. As mentioned by the reviewer, this was a typing mistake when moving the data from the datasheet to the manuscript. The calculations were repeated and corrected in the revised manuscript as recommended by the reviewer. Fortunately, this typing mistake did not affect any of the discussions or conclusions in the manuscript.

2. Page 4, line 28-29: The Authors state that "That very old method [6] included the determination of DAN only in biological fluids". It is not a drawback of the method of being published at 1973. Meanwhile, the same technique is applied in the current method. In addition, the reported spectrofluorimetric method is superior compared to the current method of being applied to biological fluids. In such analysis, sample clean up procedure should be performed to extract the drug from the complex matrix of the biological sample and this is not considered a drawback of the method.

Response:

To be extra careful and to avoid any misunderstanding, we decided to remove this sentence in the revised manuscript. In the original manuscript, the authors did not intend to show that there was a flaw in the 1973 method but only to show the differences between our proposed method and the 1973 method.

3. Page 8, line 10: the Authors state that " Since DAN on its own has no native fluorescence (Fig. 5a), the proposed spectrofluorometric " This statement is not correct because the reported method in reference [6] was based on measuring the native fluorescence of the drug. This sentence should be modified.

Response:

Practically, DAN has no native fluorescence at the emission wavelength used for the reduced fluorophore (279/344) and this is illustrated in Figure 4. However, this sentence was modified in the revised manuscript as recommended by the reviewer.

4. Page 11, section 4.3. Greenness evaluation: The authors should summarize this section and delete the well-known details.

Response:

Section 4.3. was summarized in the revised manuscript as recommended by the reviewer.

5. Page 14, line 10: what is "RCNP

Response:

We thank the reviewer for pointing out this typo to us, which was corrected to "RDAN" in the revised manuscript.

6. The structure of the studied drug presented in Figure 1, page 15, is repeated in scheme 1. So Figure 1 can be deleted.

Response:

Figure 1 was deleted and was replaced inside the text by scheme 1 in the revised manuscript, as recommended by the reviewer.

7. Figure 5a in page 17 its title is "Figure 5a: Native fluorescence spectra of DAN (1.0 $\mu\text{g mL}^{-1}$). " However, the figure shows only excitation spectra of the drug as the band is around 279 nm . The emission band is not appeared at all. The authors should present for curves; two excitation spectra of the drug and its reduced form and two emission spectra for he drug and its reduced form. The concentration of the drug and its reduced form should be the same . This will show the effect of reduction on the fluorescence spectra.

Response:

Figure 5a does not show the excitation spectrum of the drug. However, the band around 279 nm corresponds to the "cut off" of the solvent used at the used excitation wavelength. It was supposed to show that there is very low or no emission at 342 nm when the excitation wavelength is 279 nm. In the revised manuscript, Figure 5a was included in Figure 4, which includes the excitation and the emission spectra of the drug and its reduced form $1.0 \mu\text{g mL}^{-1}$ each, as recommended by the reviewer.

8. Figure 5b in page 17 has no meaning because the third dimension is not clear (The vertical axis is for RFI values and the horizontal is for wavelength, So the third (perpendicular to the page) is standing for what??).

Response:

Figure 5b was replaced by figure 4 which is a 2D figure in the revised manuscript, as recommended by the reviewer.

9. Figure 7 at page 18: There are 4 bars at pH 6.5, the blue bar is stand for the reduced drug without any buffer, However, the pH of this solution would be ≈ 2.0 but not 6.5 due to the presence of the remaining HCl from the reduction step. Acetate buffer cannot be used outside the pH range of 3.7 – 5.7 otherwise, it will lose its buffer action.

Response:

The pH on the abscissa represents the pH of the added solution (diluting solvent, which is bidistilled water or buffers), not the pH of the final solution. To avoid confusion and to show that there is no effect of adding buffers, the blue bar of the reduced drug without any buffer was moved to the beginning of the chart.

The effect of the acetate buffer outside the pH range of 3.7-5.7 was eliminated from the revised manuscript.

10. Page 21, table 2, the values of S.D for both slope and intercept are extremely low and should be re-calculated. Although, the method involved multi-steps including; reduction for 20 minute, filtration and completion with the solvent. These step should elevate the values of SD,

Response:

The authors thank the reviewer for pointing out that typing mistake. This the same mistake related to comment 1 of the same reviewer.

As mentioned in comment 1, there was a mistake in moving the data from the datasheet to the manuscript. The calculations were repeated and corrected in the revised manuscript, as recommended by the reviewer.

11. Table 1 page 20: The text in the last column should be deleted and moved to the manuscript. The table should include only numerical values.

Response:

The authors would prefer to keep the last column in Table 1, as we believe that presenting the comparison in a table form clearly shows the differences between the reported and the proposed methods.

12. Tables should be summarized and individual data should be deleted.

Response:

The tables were summarized, as recommended by the reviewer and the detailed tables were attached as a supplementary file (Supplementary 1).